# Long Noncoding RNAs and Cancer Stem Cells: Dangerous Liaisons Managing Cancer

**DOI:** 10.3390/ijms24031828

**Published:** 2023-01-17

**Authors:** Silvia Anna Ciafrè, Monia Russo, Alessandro Michienzi, Silvia Galardi

**Affiliations:** 1Department of Biomedicine and Prevention, University of Rome Tor Vergata, 00133 Rome, Italy; 2Department of Biology, University of Rome Tor Vergata, 00133 Rome, Italy

**Keywords:** lncRNAs, ceRNAs, RNA-binding protein, cancer stem cells, epigenetic regulation

## Abstract

Decades of research have investigated the mechanisms that lead to the origin of cancer, striving to identify tumor-initiating cells. These cells, also known as cancer stem cells, are characterized by the ability to self-renew, to give rise to differentiated tumor populations, and on a larger scale, are deemed responsible not only for tumor initiation but also for recurrent tumors, often resistant to chemotherapy and radiotherapy. Long noncoding RNAs are RNA molecules longer than 200 nt, lacking the ability to code for proteins, with recognized roles as fine regulators of gene expression. They can exert these functions through a variety of mechanisms, acting at almost all steps of gene expression, from modulation of the epigenetic state of chromatin to modulation of protein stability. In all cases, lncRNAs do not work alone, but they always interact with other RNA molecules, either coding or non-coding, or with protein factors. In this review, we summarize the latest results obtained about the involvement of lncRNAs in the initiating cells of several types of tumors, and highlight the different mechanisms through which they work, while discussing how the modulation of a lncRNA can affect several aspects of tumor onset and progression.

## 1. Introduction

A major challenge in cancer therapy is the fact that not all cells within a tumor are equivalently sensitive to most therapies. This is due to an intrinsic cellular heterogeneity, primarily responsible for the resistance to medical treatments and relapses that are commonly seen in malignant tumors. The heterogeneity can be explained by the cancer stem cell (CSC) or tumor-initiating cell (TIC) hypothesis, which refers to a specific tumor cell population, capable of driving tumor growth and recurrence, and increasing DNA repair resistance [1]. CSCs were first identified in acute myeloid leukemia (AML); the study demonstrated that AML CSCs with the phenotype CD34^+^ CD38^−^ constituting approximately 0.1–1% of the tumor population, can generate AML in mice [2]. AML CSCs were revealed to share many of the hallmarks of normal stem cells, including the ability to self-renew and differentiate into other cell types [2]. Further studies have shown that CSCs play a prominent role in other hematological and solid cancers; they influence the resistance to cancer therapies such as chemotherapy and radiation therapy [3]. Moreover, CSCs have an increased metastatic potential due also to their ability to undergo the epithelial-to-mesenchymal transition (EMT). When CSCs initiate EMT programs the resulting cells become invasive, acquire a migratory behavior, and are responsible of the formation of distal tumors [4]. In healthy tissue, stem cells reside in a “stem cell niche”, a specific microenvironment that plays a strategic role in regulating the maintenance and self-renewal of stem cells through direct cell-to-cell contact that interferes with pathways of self-renewal and differentiation. An analogous concept can be applied to CSCs where there is also a cancer-specific “cancer stem cell niche”, and interactions with this niche are essential to drive tumor growth through interaction with resident and infiltrating non-staminal tumor cells [5]. Studies on cancer stem cell niches have shown that tumor-specific microenvironments comprise a variety of cells, such as stromal, immune, and epithelial cells. In addition, a network of extracellular macromolecules supports cells within the extracellular matrix, significantly promoting stemness maintenance, invasion, and formation of metastases [6]. These environmental factors contribute to aberrantly activate multiple signaling pathways, such as Wnt, Hedgehog, and Notch, which are responsible for stemness and differentiation and are normally involved in embryogenesis and homeostasis [6]. They also influence the regulation of gene expression of transcription factors such as SOX2, NANOG, OCT4, KLF4, and c-MYC, which have been shown to be crucial for CSCs to maintain their self-renewal capacity [7]. While in some cancers gene-encoding members of stem cell signaling pathways are recurrently mutated, in other cancers these same genes are often de-regulated epigenetically. For instance, NOTCH1 is epigenetically activated in breast CSCs [8], as is WNT signaling in leukemias [9]. Defined stem cell signals, such as the protein Musashi homologue (MSI) have also been shown to be both genetically and epigenetically modified in cancers; for example, MSI2 translocation is associated with the accelerated phase in chronic myeloid leukemia, but MSI2 can also be epigenetically activated in the absence of mutations [10]. Recent studies suggested that epigenetic mechanisms are a driving force contributing to phenotypic differences between CSCs and other tumor cells [11,12]. In addition to epigenetic modifications, post-transcriptional RNA processing alterations, which cause the recoding of transcripts or alter translational efficiency, have been shown to influence stem-cell function in malignant settings. Recently, several groups have identified master post-transcriptional regulators of CSC genetic programs, including RNA modifications, RNA-binding proteins, noncoding RNAs, that can be considered predictive biomarkers of stem cell fitness. Of particular interest, these studies reveal that different post-transcriptional mechanisms are coordinated to control key signaling pathways and transcription factors to either support or suppress CSC homeostasis [13]. So, aberrations due to epigenetic and post-transcriptional modifications can play a key role in generating intra-tumor heterogeneity and further complicating the therapy. A deeper understanding of how these mechanisms contribute to the acquisition or maintenance of a stem cell phenotype is critical for developing strategies to disable the early permissive states for effective prompt intervention or prevention strategies. Several authors have previously reviewed the role of long noncoding RNAs (lncRNAs) in cancer, often focusing on specific features, such as the signaling pathways driving cancer stemness [14], or the epithelial-to-mesenchymal transition [15]. Here, we specifically focus our review on the role of lncRNAs as regulators of the epigenetic and post-transcriptional state of CSCs. We provide and discuss relevant findings published in the last five years for an updated review of lncRNAs that play oncogenic or tumor suppressor roles. From a mechanistic point of view, we discuss examples of lncRNAs in cancer stem cells by distinguishing them based on their subcellular localization and, consequently, their mode of action.

## 2. Long Non-Coding RNAs

Genomes are extensively transcribed and give rise to thousands of lncRNAs, which are defined as a class of RNAs with transcripts longer than 200 nucleotides (nt), with limited or no protein-coding capacity [16]. Countless studies have already revealed that lncRNAs are relevant to many physiological and pathophysiological processes, including embryonic development, organ formation, and tumorigenesis [17]. Previous studies have commonly concentrated on the effects of the protein-coding genome on cancer stemness. However, over the past several decades, understanding of the non-coding genome and its influence on cell fates has expanded greatly [18]. LncRNAs are now recognized as a new potential area of cancer therapy as they play essential roles in regulating cancer cell proliferation, chemoresistance, and metastasis.

### 2.1. Biogenesis and Conservation of lncRNAs

The biogenesis of lncRNAs resembles that of messenger RNAs (mRNAs). They are frequently transcribed using RNA Polymerase II (Pol II). The resulting lncRNAs are often capped by 7-methyl guanosine (m^7^G) at their 5’ end, polyadenylated at their 3’ end, and spliced. They also contain classical splice sites (GU/AG), exhibit alternative splicing, and their genes are associated with the same histone modifications as protein-coding genes [19]. However, compared to their protein-coding counterparts, lncRNAs are composed of fewer exons, are subject to weaker selective constraints during evolution, and are expressed at a relatively lower abundance. The reason for this difference is unclear, although it may reflect the functional specificity of coding for protein versus regulating expression [20].

In addition, lncRNA expression is strikingly cell- and tissue-specific and, in many cases, even primate-specific [21].

### 2.2. LncRNAs Classification

Based on their localization in the genome, compared to protein-coding genes, lncRNAs genes and their transcribed products can be divided into various subclasses:-sense lncRNAs and antisense lncRNAs overlap with one or more exons of neighboring mRNAs, respectively on the same and the opposite strand;-intronic lncRNAs are transcribed from introns of the protein coding genes;-promoter upstream *lncRNAs* or promoter-associated lncRNAs whose genes are located upstream or close to the promoter of a different gene;-intergenic lncRNAs, whose genes lie within the genomic interval between two genes;-bidirectional lncRNAs, which share the promoter with protein-encoding genes but are transcribed in the opposite direction;-3’ UTR-associated RNAs, derived from the 3’-untranslated region of the protein-coding transcript [22].

Each of these genomic arrangements can produce physiologically relevant lncRNAs.

Another approach to classify lncRNAs is based on their subcellular localization, which is closely related to their biological function and potential molecular roles. Compared to mRNAs, a more significant portion of lncRNAs are localized in the nucleus, raising the fundamental question of what drives their differential localization. Overall, lncRNAs are spliced less efficiently than mRNAs. They have weaker splicing signals and longer distances between the 3’ splice site and the branch point that correlate with augmented retention. Other factors, such as the differential expression of certain splicing regulators, also contribute to the accumulation of lncRNAs in the nucleus [23]. In the nuclei, lncRNAs are associated with chromatin organization, transcription, and RNA processing. They can regulate transcription by influencing transcriptional factor activity, assembling Pol II machinery, and interacting with chromatin-modulating proteins such as subunits of SWI/SNF or polycomb repressive complexes (PRC) [24]. At the epigenetic level, lncRNAs can recruit a variety of proteins, can regulate histone activity through acetylation, methylation, and ubiquitination, and are directly related to signaling mediators, such as receptors, protein kinases, and transcription factors, and regulate their enzyme activities [25].

On the basis of the mode of action, lncRNAs can be divided into *cis*-acting lncRNAs and *trans*-acting lncRNAs. The genomic position of *cis*-acting lncRNAs should be sufficiently close to the genes they regulate. These lncRNAs are located upstream or downstream of a protein-coding gene and modulate local gene expression at the transcriptional level by recruiting transcription factors and/or chromatin modifiers [26]. They could also form a DNA–RNA triplex that anchors the lncRNA and the effector proteins associated with the gene promoter. These types of lncRNAs usually depend on their sequences to regulate the expression of neighboring genes [27].

The lncRNAs that act in *trans* need to be relocated from their synthesis site to play their regulatory roles and influence gene regulation, interacting with transcription factors (TFs), chromatin modifiers, or other regulatory elements.

In the cytoplasm, lncRNAs can play regulatory roles mediating RNA processing, affect mRNA stability, or directly regulate protein function. mRNA regulation can be carried out through various mechanisms: they can improve mRNA stability by directly binding to the 3’-untranslated regions (UTR) of target genes [28], or regulate mRNA stability indirectly by interacting with RNA-binding proteins (RBPs). These RBPs function as chromatin modifiers, transcription factors, and regulators of their neighboring protein-coding genes [29].

Multiple reports show that one of the most frequent activities of the cytoplasmic lncRNAs is to act as decoys for miRNAs, short non-coding RNAs that mediate the RNA interference (RNAi) by post-transcriptional mechanisms. LncRNAs that bind miRNAs and prevent their interaction with their target are regarded as competing endogenous RNAs (ceRNAs), decoys, or sponges. Since they prevent miRNAs from fulfilling their regulatory function, lncRNAs acting as sponges are positive regulators of target mRNA transcripts [30].

Identifying this level of gene regulation can also explain the correlation between genome size and increased species complexities [31]. Most lncRNAs capture miRNAs using a region close to their 3’ end named miRNA Response Elements (MRE), which interacts with the “seed region”, 2-8 nucleotides at the 5’ end of the miRNA. This region is particularly crucial for the recognition and silencing or interaction with ncRNAs. MRE is also complementary to the Argonaute (AGO) binding site, which participates in the last step of miRNA maturation, thus generating the RNA-induced silencing complex (RISC). RISC exerts the biological function of the last authentic effector molecule, where AGO works as the catalytic engine, facilitating miRNA binding to the target site of mRNA, and cleaving the miRNA-mRNA duplex by its endonuclease activity [32].

### 2.3. LncRNAs and Organelles

Several lncRNAs are sorted into mitochondria by unknown mechanisms. Mitochondria-localized lncRNAs can be encoded by both nuclear and mitochondrial DNA and are often associated with mitochondrial metabolism, apoptosis, and the crosstalk of mitochondria with nuclei. For example, the RNA component of mitochondrial RNA-processing endoribonuclease (RMRP) is recruited to mitochondria and is stabilized by binding G-rich RNA sequence-binding factor 1 (GRSF1) [33]. Some lncRNAs are found in exosomes, probably by forming lncRNA-RBP complexes [34]. Exosomes are small membrane vesicles (30–150 nm) that are constantly secreted by cells and deliver macromolecular messages that enable cell communication. Because exosomes are regularly released into the extracellular environment, exosome-localized lncRNAs can be secreted and end up in recipient cells, where such lncRNAs are found to be involved in epigenetic regulation, cell-type reprogramming, and genomic instability [35]. The discovery of other organelle-specific lncRNAs will likely provide additional mechanistic insight into the connection between lncRNA regulation and organelle homeostasis.

## 3. Nuclear lncRNAs and CSCs

In the nuclei, a significant fraction of lncRNAs have been associated with chromatin organization, transcription, and RNA processing (Table 1).

As described above, lncRNAs can regulate transcription by influencing transcription factor activity, assembling Pol II machineries, and interacting with chromatin modulating proteins such as the ATP-dependent chromatin-remodeling complexes. The INO80 remodeling complex is a conserved complex that modifies chromatin using the energy of ATP [44,45]. The INO80 complex controls gene expression, DNA damage repair and replication [46], as well as maintaining mammalian stem cell properties [47].

Wang and colleagues showed that in liver CSCs, the INO80 complex is associated with the conserved lncRNA HAND2-antisense RNA 1 (HAND2-AS1) [36]. HAND2-AS1 is a lncRNA transcribed in divergent direction with respect to the HAND2 gene. The depletion of HAND2-AS1 transcripts by loss-of-function approaches does not affect the expression levels of its divergent protein-coding gene HAND2 and other nearby genes. Rather, HAND2-AS1 acts *in trans* by recruiting the INO80 complex to the BMPR1A (Bone Morphogenetic Protein Receptor Type 1A) to initiate its expression, leading to activation of BMP (Bone Morphogenetic Proteins) signaling and increasing the self-renewal and maintenance of liver CSCs.

In triple-negative breast cancer (TNBC), SMARCA4, a core subunit of the ATPase-dependent protein complex SWI/SNF, interacts with the lncRNA TGFB2-AS1 [37], an antisense RNA to the protein coding TGFB2 (Transforming Growth Factor Beta 2) gene. Unlike the lncRNA described above, TGFB2-AS1 counteracts CSC maintenance, and its overexpression in an orthotopic TNBC mouse model remarkably abrogates the enhancement of tumor growth and lung metastasis endowed by SWI/SNF target genes. Mechanistically, the exon 3 of TGFB2-AS1 directly interacts with the N-terminal of SMARCA4, antagonizing its binding capacity on TGFβ2 and SOX2 (sex determining region Y-box 2) promoters. TGFβ2 and SOX2 transcription repression leads to the inhibition of both CSC signaling and Transforming Growth Factor β (TGF β) signaling. TGF β signaling plays a key role in cancer progression through its effects on gene expression, the release of immunosuppressive cytokines, and epithelial plasticity, and thus enhances cancer cell invasion and dissemination, CSC properties, and therapeutic resistance [48]. The importance of the cooperative activity of TGFβ2 and CSC signal pathways emphasizes the regulatory mechanism of this lncRNA that the authors found acting *in cis* (TGFB2) and *in trans* (SOX2) in the same cellular context. To explore the mechanism that regulates the transition from CSCs that can self-renew and proliferate slowly to rapidly growing cancer cells, Zagorac and collaborators evaluated how gene expression changes during sphere formation in aggressive breast cancer [38]. They described how the two transcription factors SOX2 and EZH2 (Enhancer of zeste homolog 2) are directly involved in the transcription of cell cycle genes and in the activation of self-renewal by recognizing CpG islets in mammary CSCs. Furthermore, the authors identified a novel lncRNA called SCIRT (Stem Cell Inhibitory RNA Transcript), which is strongly upregulated during tumor sphere formation, but unexpectedly counteracts stemness. Functionally, SCIRT colocalizes with and counteracts EZH2 and SOX2 during cell-cycle and self-renewal regulation to restrain tumorigenesis. The authors showed that SCIRT interacts with EZH2 to increase EZH2 affinity to its protein partner FOXM1 (Forkhead Box M1). In this manner, SCIRT induces transcription at cell-cycle gene promoters by recruiting FOXM1 through EZH2 to antagonize EZH2-mediated effects at the target genes. Conversely, on stemness genes, FOXM1 is absent, and SCIRT antagonizes EZH2 and SOX2 activity, balancing toward repression. By acting in this negative feedback loop, SCIRT increases cell-cycle and represses the transcriptional programs of self-renewal in CSCs. This study suggests that SCIRT acts as a regulator that only reduces the transcription of genes involved in tumorigenesis without fully suppressing their activity. In this manner, SCIRT tends to be more expressed in tumors than normal cells, but when it is expressed, it is associated with a more favorable prognosis.

In another study, a lncRNA termed Low expressed in Bladder Cancer Stem cells (lnc-LBCS), which contributes to weak tumorigenesis and enhanced chemosensitivity of bladder CSCs, was shown to recruit EZH2 to exert an unexpected tumor-suppressive role [39]. Lnc-LBCS physically interacts with EZH2 and the heterogeneous nuclear ribonucleo-protein K (hnRNPK), and serves as a scaffold to induce the formation of the hnRNPK–EZH2 complex. Although hnRNPK and EZH2 are overexpressed and exert oncogenic functions in several cancers, recent studies indicated that some lncRNAs could convert them into tumor-suppressors [49,50]. This study showed that lnc-LBCS indeed plays this convertor role; it recruits the hnRNPK–EZH2 complex to the SOX2 promoter and suppresses SOX2 expression by mediating the tri-methylation of lysine 27 in the histone H3 protein. Furthermore, the study showed that SOX2 is required for the self-renewal and chemoresistance of bladder CSCs and rescues the suppressive effect of lnc-LBCS.

An intriguing mechanism of action of lncRNAs involves their interaction with RBPs that work in N^6^-methyladenosine (m^6^A) mRNA modification, either as readers or as erasers of this post-transcriptional modification [51]. A pivotal work describing such a relationship in glioblastoma stem-like cells (GSCs) demonstrated that the lncRNA FOXM1-AS promotes the interaction of FOXM1 nascent transcripts with the m^6^A eraser ALKBH5 (Human AlkB homolog H5), thus enhancing the demethylation of FOXM1 nascent transcripts and ultimately FOXM1 expression [40]. The downstream effects of FOXM1 modulation are widespread, as this transcription factor is a known master regulator of the cell cycle, with a role in the establishment, maintenance, and functions of CSCs [52]. Notably, these authors showed that the depletion of FOXM1-AS, as well as of ALKBH5, disrupted GSC tumorigenesis through the FOXM1 axis. This suggests that FOXM1-AS could be investigated as a novel target to fight the tumor-initiating cells in glioblastoma and possibly in other cancers.

What is common to many of the recent works which dig deeply into lncRNA-mediated mechanisms is that they describe key functions for lncRNAs by filling knowledge gaps in pathways previously depicted but still showing some dark areas. This is the case of Ji J. and collaborators’ work [41], where the lncRNA SChLAP1 (Second Chromosome Locus Associated with Prostate-1) is shown to bind to HNRNPL (Heterogeneous Nuclear Ribonucleoprotein L) in glioblastoma cells, both stem, and non-stem; the formation of the protein–RNA complex stabilizes SChLAP1 and promotes the interaction between HNRNPL and ACTN4 (Actinin Alpha 4), in turn increasing the stability of the ACTN4 protein by inhibiting its ubiquitination and proteasomal degradation. Building on the known activity of ACTN4 as a transcriptional coactivator of the RelA/p65 subunit of Nuclear factor-κB [53], Ji et al. could demonstrate that the SChLAP1–HNRNPL complex indeed promoted transcriptional activity and nuclear translocation of Nuclear factor-κB via enhancing ACTN4 stability in glioblastoma cells. From a translational perspective, a promising aspect of this work is that the authors defined the specific protein domains through which HNRNPL interacts with SChLAP1, and thus commented that this might provide the theoretical basis for the selection of drugs working by interfering with these interactions.

To define therapeutically targetable lncRNAs involved in the maintenance of medulloblastoma (MB) cells with a stem-like phenotype, Peng and collaborators analyzed the role of lncRNA MIAT (Myocardial Infarction Associated Transcript) that they found highly expressed in the Sonic Hedgehog group of MB and is required for the preservation of CSCs in the disease [42]. MIAT was initially identified as a myocardial infarction-associated transcript [54] that has more recently been found to be especially enriched in neural tissues and has been associated with various diseases, including schizophrenia and Parkinson’s disease. The loss of MIAT enforces the differentiation of MB CSCs into a non-tumorigenic state. MIAT expression also facilitates resistance to treatment by down-regulating p53 signaling and altering radiation-induced cell death. Authors found that MIAT interacts with the protein Metadherin, but how these two molecules cooperate and the importance of this interaction to their mechanism in MB cells is still under investigation. Metadherin is an RNA-binding protein previously shown to contribute to mechanisms of chemoresistance and is required for the tumor-initiating capacity of cancer cells [55]. Since experiments on Metadherin loss and MIAT promoter knockouts showed similar effects on the transcription of a subset of microRNAs, the authors speculated that MIAT and Metadherin could function in a common mechanism to regulate microRNA biogenesis, revealing a novel way through which a lncRNA can interfere with microRNA functions, rather than simply by competing for their binding as will be described in the next paragraph.

## 4. Cytoplasmic lncRNAs and CSCs

### 4.1. Long Noncoding RNAs Working as Competing Endogenous RNAs

The role of lncRNAs acting as ceRNAs was one of the first to be discovered in physiological and pathological settings. As described above, many works have shown that lncRNAs work through this mechanism, where they act as ‘natural sponges’ for microRNAs, thus subtracting microRNAs from their physiological inhibitory function. Due to the pervasive effect of microRNAs, essentially all key aspects of cancer cells, and in particular of cancer stem cells, can be affected by this ceRNA function of lncRNAs (Table 2).

In breast cancer, Peng F. and co-authors provided a picture of the lncRNA H19 as deeply embedded in a signalling axis needed to reprogram breast cancer stem cells under hypoxia [56]. H19 sponges microRNA let-7 to release HIF1-α (Hypoxia-inducible factor 1α), which in turn activates a cascade of target genes, among which the glycolytic enzyme PDK1 (Phosphoinositide-dependent kinase-1) plays a master role in breast cancer stem cells. They show that, under hypoxia, PDK1 is needed to reprogram breast cancer cells toward stemness. This program is triggered and sustained by H19, whose high levels in hypoxic tumor regions sequester let-7, leaving the mRNA for HIF1-α and other targets, such as DICER, available for translation.

The hypoxic environment is key to the function of lncRNA RUNX1-IT1 too, which, opposite to H19, works as a tumor suppressor in hepatocellular carcinoma. Sun et al. [57] showed that this intronic lncRNA is repressed under hypoxia using HDAC3 (Hypoxia-Driven Histone Deacetylase 3). Thus, the decrease in RUNX1-IT1 releases the “oncomiR” miR-632, which in turn represses GSK-3β (Glycogen Synthase Kinase-3 beta), which subsequently induces the Wnt/β-catenin pathway. In this way, RUNX1-IT1 downregulation is a key effect of the hypoxic microenvironment, and widely affects HCC, increasing stemness and the tightly connected epithelial-mesenchymal transition (EMT), invasiveness, and metastatic ability.

He et al. provided a further example of how a ceRNA-miRNA relationship can affect the Wnt pathway and the stemness in tumor cells. In non-small-cell lung cancer (NSCLC), they found that the lncRNA PKMYT1AR sequesters the tumor suppressor miR-485-5p, thus determining the efficient translation of the oncogenic protein kinase, membrane-associated tyrosine/threonine 1 PKMYT1 [58]. This, in turn, constitutively activates the Wnt pathway because it blocks the binding of the E3-ligase SCF^β-TrCP^ to β-catenin and reduces its degradation. In view of PKMYT1AR function in NSCLC, the authors also targeted it in vitro and in tumor xenografts, showing that its inhibition through ASOs can efficiently impair tumor stem features, as self-renewal, and the expression of stem cell markers, as CD44 and SOX2. Furthermore, the intratumoral injection of ASOs targeting this lncRNA induced a reduction in tumor growth.

An additional lncRNA, SOX2-OT (SOX2 Overlapping Transcript), known to play an oncogenic role in several cancer types [68], was demonstrated to exert its function working as a ceRNA in bladder cancer, where, by sponging miR-200c, regulates the stem cell master transcription factor SOX2 [59]. The consequent high expression of SOX2 enhances many stemness hallmarks of bladder cancer cells in vitro, and increases the tumorigenic potential and, most notably, the metastatic ability of tumor cells in vivo.

In melanoma, a recent paper unveiled a regulatory circuit centered on the lncRNA LHFPL3-AS1 (LHFPL3 Antisense RNA 1): by sequestering miR-181a-5p, it positively affects the expression of BCL2 (*B-cell lymphoma 2*), thus inhibiting the apoptosis of melanoma cells, and in particular melanoma stem cells [60]. An interesting aspect of this work is that the authors found that only one of the two annotated splicing isoforms of LHFPL3-AS1 works as a ceRNA in this circuit. The longer one, named LHFPL3-AS1-long, includes exon three but does not contain exon four, present in the shorter isoform, and harbours the miR-181a-5p site needed for its ceRNA function. The intriguing results obtained by Zhang and co-authors showed that the alternative splicing of the LHFPL3-AS1 precursor in melanoma stem cells is mediated by the PTBP1 (Polypyrimidine Tract Binding Protein 1) which, by binding to a specific motif (UCUCU) in exon three of the LHFPL3-AS1 precursor, favours its inclusion in the mature transcript. As a whole, this work provides an example of a complex network of regulatory factors—lncRNA, miRNA, mRNA, splicing factor, and splicing isoforms—cooperating to maintain the aggressive phenotype of melanoma stem cells and suggests that interfering with the discovered nodes of this network could help to fight this cancer.

Two articles recently linked the activity of two different lncRNAs that act as ceRNAs in gastric cancer to fatty-acid oxidation (FAO) and stemness and chemoresistance [61,62]. In fact, FAO is a major pathway that regulates fatty-acid degradation and promotes the production of ATP and NADPH, whose perturbation is tightly linked to tumor progression [69], stemness, and chemoresistance [70]. A key aspect shared by both works is the focus on the involvement of mesenchymal stem cells (MSC) in the acquisition of chemoresistance by tumor cells, coupled with increased stemness of cancer cells. Thus, the perspective of these two papers is widened to encompass not only tumor cells, but also the microenvironment, represented, in gastric cancer, by a large number of MSCs. Both papers describe lncRNAs whose expression is induced in gastric cancer cells by the co-culture with mesenchymal stem cells, and is directly correlated to stemness and chemoresistance, and both works identify a specific microRNA that is sponged via the lncRNA. He and co-authors found that the lncRNA MACC1-AS1 (MACC1 Antisense RNA 1) is stimulated in gastric cancer cells in part by the TGFβ1 secreted by MSCs, acts by sequestering miR-145-5p and, by doing so, induces the expression of the FAO enzymes CPT1 (carnitine palmitoyltransferase 1) and ACS (acetyl-coenzyme A synthetase) [61]. However, the authors could not find any direct binding sites for miR-145-5p on the CPT1 and ACS mRNAs. This suggests that an intermediate factor, encoded via an mRNA yet to be defined, is the direct target of miR-145-5p and is made available for efficient translation in the presence of high levels of MACC1-AS1.

The other work, by Wu et al., focuses on HCP5 (HLA Complex P5), a lncRNA induced by co-culturing gastric cancer cells with MSCs that sponges miR-3619-5p and, as for MACC1-AS1, boosts FAO through increased CPT1 expression and activity [62]. This, in turn, increases the stemness, aggressiveness, and chemoresistance of gastric cancer cells in vitro and in vivo. These authors take an additional step toward understanding the mechanism linking the lncRNA, its related miRNA, and the increase in FAO and CPT1. In fact, they identify the direct target of mir-3619-5p, whose repression is released when HCP5 is highly expressed. They show that PPARGC1A mRNA, coding for the transcriptional co-activator PGC1α deeply involved in FAO [71], is repressed by mir-3619-5p. This reduces the active transcriptional complex formed by PGC1α and CEBPB (CCAAT Enhancer Binding Protein Beta), needed for the transcriptional activation of CPT1 [72]. These findings complete the puzzle of the HCP5 molecular mechanism in gastric cancer cells. It would now be intriguing to verify if mir-145-5p, identified by He et al. [61] as sponged by MACC1-AS1 in the same cells upon stimulation by MCS co-culture, and leading to the activation of the same FAO enzymes, also targets PPARGC1A mRNA to affect CPT1 transcription.

All these recent findings highlight that the contribution of lncRNAs in their ceRNA mode-of-action is relevant to all types of cancer, works both directly in tumor cells and indirectly, with the mediation of cells of the microenvironment, and affects the expression of key genes and pathways of cancer stem cells at all levels.

### 4.2. Long Noncoding RNAs Working through the Interaction with RNA-Binding Proteins

LncRNAs can bind to RBPs and variably modulate their functions, usually determining a wide range of downstream effects. Indeed, several recent works have shown that the binding of a cytoplasmic lncRNA to an RBP can profoundly impact cancer stem cell biology, contributing a further step in understanding stemness in cancer, tightly linked to resistance to therapy (Table 2).

In breast cancer, a novel lncRNA induced by the hypoxic environment, named KB-1980E6.3, was found as a key factor for the maintenance of cancer stemness, where it plays its role by interacting with an important player of the m^6^A machinery, the reader IGF2BP1 (Insulin Like Growth Factor 2 mRNA Binding Protein 1) [63]. Binding of KB-1980E6.3, in turn, enhances the recognition of m^6^A-modified c-Myc mRNA by IGF2BP1, and hence the stability of c-Myc mRNA, thus triggering a cascade of oncogenic events, resulting in breast cancer cell stemness and tumorigenesis, both in vitro and in vivo.

An intricate mechanism links the DDIT4-AS1 lncRNA to the m^6^A eraser ALKBH5, the RBP HuR (Human antigen R), the stability of the DDIT4 (DNA Damage Inducible Transcript 4) mRNA, and the mTOR pathway in pancreatic cancer stem cells [64]. This lncRNA, highly expressed in pancreatic carcinoma, is itself a target of m^6^A modification and is stabilized by binding the of RBP HuR to m^6^A sites. High levels of DDIT4-AS1 allow its physical interaction with UPF1 and the promotion of UPF1 phosphorylation due to the inhibition of SMG5 and PP2A binding. Phosphorylated UPF1 then drives the degradation of the DDIT4 mRNA (to which DDIT4-AS1 is the antisense) and the activation of the mTOR pathway on the basis of stemness and chemosensitivity of pancreatic carcinoma cells [73]. Therefore, in this context, in line with what was observed in glioblastoma stem cells [40], the functional interaction between an antisense lncRNA and its “sense” mRNA is a key event, but, contrary to the FOXM1-FOXM1-AS case described in the nuclear lncRNA paragraph, DDIT4-AS1, antisense to DDIT4, induces the downregulation of DDIT4 mRNA.

Liu Y. et al., by reporting the mechanism of action of the lncRNA NR2F1-AS1 (NAS1), also explore the intricate definition of cancer stem cells in breast cancer [65]. They find that NAS1, on the one hand, maintains the ability of tumors to invade locally, but on the other hand, its expression results in the formation of more long-term solitary tumor cell foci and fewer metastatic nodules in the lungs. Ultimately, the authors show that NAS1 can promote the dormancy of breast cancer metastatic cells, and it does so by binding to the GC-rich 5′UTR of NR2F1 (Nuclear Receptor Subfamily 2 Group F Member 1) mRNA, recruiting the RBP PTBP1, which enhances the internal ribosome entry site-dependent translation of NR2F1, a transcription factor known to promote tumor dormancy in breast cancer [74]. The downstream effect of NR2F1 increased production is then the suppression of the transcription factor and marker of epithelial stem cells ΔNp63 [75] expression, leading to cellular EMT changes and impaired tumorigenicity. Notably, the data reported by this work integrate the definition of breast cancer stem cells as a heterogeneous population, composed of mesenchymal-like CD24^−^CD44^high^ cells, prone to dissemination and dormancy, characterized by a higher NAS1 expression, and of epithelial-like ALDH^+^ cells, metastasis-competent and representing awakening from dormancy, and more tumorigenic than the mesenchymal-like stem cells. As a supporting finding with clear relevance to clinics, this work showed that NAS1 was expressed more in late-recurring tumors than in those that relapsed within the first 2 years after diagnosis.

Very recently, a work has been published which investigates not only the lncRNAs enriched in stem-like and chemo-resistant glioblastoma cells, but also those whose expression is affected by the selective inhibition of HDAC6 (Histone Deacetylase 6) through azaindolylsulfonamide (MPT0B291) [66]. This drug is known to induce cellular senescence in stem-like GBM cells and to prolong survival in mouse models of temozolomide-resistant xenografts through the downregulation of Sp1 and its target genes associated with drug resistance [76]. Wu and collaborators report that the inhibition of HDAC6 by either siRNAs or MPT0B291 results in the suppression of LINC00461, which, however, is not dependent on Sp1. Rather, LINC00461 was stabilized by the interaction between HDAC6 and RNA-binding proteins (RBPs) such as CNOT6, a typical protein related to mRNA decay [77]. In the model proposed by the authors, HDAC6 deacetylase activity is needed to deacetylate both CNOT6 (CCR4-NOT Transcription Complex Subunit 6), thus leading to the suppression of its RNA decay activity, and FUS, thus, in contrast, boosting its ability as a stability enhancer on LINC00461. These authors only partially investigated the mechanism downstream LINC00461 stabilization and showed that this lncRNA can work as a ceRNA for miR-485-3p, a recognized tumor-suppressor microRNA in glioblastoma [78], but these results were only obtained in one non-stem cell line of glioblastoma, and more studies are needed to unravel how this lncRNA affects the stemness of glioblastoma cells.

The message provided by all the works described here and many others is that lncRNAs can interact with RNA-binding proteins and can thus either be modified by them or modulate RBP functions and targets. This suggests that future research on the functional RBP–lncRNA relationship will predictably find more examples and uncover the mechanistic details based on its role in cancer initiation and progression.

## 5. Exosomal lncRNAs

Exosomes are small (30 to 150 nm) membranous vesicles originating from multivesicular endosomes that can transfer some intracellular cargoes, including lncRNAs, between cells [79]. A more recent focus has been on exosomes that play a role in transmitting information between CSCs and non-CSCs, resulting in the activation of CSCs for cancer progression and the modulation of their surrounding microenvironment. Emerging evidence showed that the molecular crosstalk between CSCs and non-CSCs in the tumor microenvironment plays a critical role in the dynamic equilibrium between CSCs and non-CSCs [80]. Exosomal lncRNAs can be considered crucial factors that mediate extracellular communication in the tumor microenvironment [81]. This is the case for DOCK9-AS2 (DOCK9 antisense RNA2), which is found to be particularly enriched in exosomes derived from CSCs of papillary thyroid carcinoma (PTC) [43]. PTC-CSCs transmitted exosomal DOCK9-AS2 to improve the stemness of PTC cells. Interestingly, DOCK9-AS2 is localized in both the nucleus and cytoplasm of PTC cells, where it positively regulates Wnt/β-catenin activity by increasing β-catenin level. In the nucleus, the lncRNA can interact with Sp1, a well-known oncogenic transcription factor in cancer [82], to activate CTNNB1, the coding gene of β-catenin (Table 1). In the cytoplasm, DOCK9-AS2 regulates CTNNB1 by acting as a ceRNA for miR-1972 (Table 2), previously found as a tumor suppressor in osteosarcoma [83]. Furthermore, the inhibition of miR-1972 plus Sp1 overexpression fully rescued Wnt/β-catenin signaling activity in DOCK9-AS2-silenced PTC cells, restoring proliferation, migration, invasion, EMT, and stemness of PTC cells. The importance of exosomal lncRNAs in the CSCs-non-CSCs dynamic equilibrium was also shown by Wei Li and collaborators [67]. The authors demonstrated that the X-linked lncRNA FMR1-AS1 (FMR1 Antisense RNA 1) was significantly overexpressed in CSCs within female ESCC (Esophageal squamous cell carcinoma) patients, and its overexpression can correlate with a poor clinical outcome. ChIRP-MS (Chromatin Isolation by RNA Purification-Mass Spectrometry) data indicated that FMR1-AS1 could be selectively packaged into exosomes derived from these cells. Functional studies demonstrated that FMR1-AS1 could interact with the Toll-like Receptor 7, TLR7 (Table 2). TLR7 was identified as one of the Pattern Recognition Receptors sensitive to exosomes that contributes to the stimulation of downstream pathways in breast cancer cells [84]. In ESCC cells, the interaction between FMR1-AS1 and TLR7 activates downstream TLR7-NFκB signaling, promoting c-Myc expression, thus inducing proliferation and invasion capacity of ESCC cells and inhibiting apoptosis. Furthermore, exosome incubation and co-xenograft assays indicated that FMR1-AS1 exosomes may be secreted from CSCs of ESCC, transferring stemness phenotypes to a non-CSC recipient in the tumor microenvironment.

## 6. Conclusions and Future Perspectives

Decades of research have been dedicated to studying the molecular and cellular basis of cancer, and great steps have been taken toward understanding it. This is translated into specific, even personalized, therapeutic approaches which would have never been possible in the past. The identification of cancer stem cells is one of such great steps. Still, in the intricate pathways leading to tumor transformation, some fine-tuning mechanisms need to be defined to fully comprehend, and possibly target, the nature of tumor-initiating cells. A wealth of work has been showing that lncRNAs can be one of the keys to subtly dissecting these pathways and completing our observation of them. These differ from protein-coding RNAs in several ways that need to be considered in the analysis of their effects, as they can play key regulatory roles in gene expression at absolute expression levels that are low compared to protein-coding genes and demonstrate tissue or cell-type specificity in expression. Furthermore, as can be seen from the examples described in this review, the effects of lncRNAs can result from very different mechanisms of action and can also participate in intercellular communication (Figure 1).

Recognizing the multiple roles these molecules can play in cancer onset and progression, we may now envisage targeting them for specific-cancer therapies. The cell-type specific expression and functions of several lncRNAs involved in cancer suggest that, by targeting these molecules, we might disrupt oncogenic pathways without affecting other components of the same pathways, that are more widely expressed also in healthy cells. Although we are still far from being able to systemically target lncRNAs and no lncRNA-targeting therapeutics have entered clinical development so far, many efforts are currently being made to increase the efficiency, specificity, and bioavailability of molecular tools aimed at disrupting their structures and functions. The latest results published in this field have all been obtained in preclinical mouse models and depict attempts to either knock down oncogenic lncRNAs or ectopically express tumor-suppressor ones. As a prototype example, Zhou M et al. showed that the chitosan gelatin nanoparticle-mediated TMEM44-AS1 silencing enhanced 5-fluorouracil toxicity in vivo in gastric cancer [85]. This was obtained by both the direct intratumor and the tail-vein injection of the lncRNA siRNA nanoparticles, in both cases showing good biocompatibility. The opposite approach is described in the work by Mao W. et al., where the lncRNA SLERCC is ectopically expressed in renal cell carcinoma via the nanoparticle-mediated delivery of a plasmid encoding the lncRNA [86]. Intravenous treatment with these nanoparticles was able to inhibit renal cell carcinoma progression in subcutaneous xenografts and also reduce metastatic nodules in tail vein metastasis models. These effects were further enhanced when the SLERCC nanoparticle was administered in combination with the tyrosine kinase inhibitor Sunitinib. In the different context of neuroblastoma, Wang and co-authors have shown that the intravenous injection of Gapmers targeting the N-Myc-responsive lncRNA MILIP, in combination with the DNA damaging chemotherapeutic drug Cisplatin, inhibits neuroblastoma xenografts more than Cisplatin or Gapmer single therapy [87]. As in these works, in most cases the modulation of the chosen target lncRNA is shown to cooperate with specific chemotherapeutic agents to improve the therapeutic effect. This may provide a chance to overcome the scarce chemotherapeutic response shown by tumors, and to impair their ability to develop resistance, tightly linked to cancer stem-cell function.

In addition to chemotherapy, immunotherapy represents a promising approach for many types of tumors [88]. However, the definition of the infiltration level of immune cells and therefore the predicted response outcomes of immunotherapy differ in tumors and in patients. Quan Cheng’s group recently published two papers describing lncRNA signatures that can be exploited as predictive biomarkers, allowing a precise selection of either the glioblastoma or low-grade glioma who would benefit from immunotherapy [89,90].

In conclusion, even if direct modulation of lncRNAs is not yet a present therapeutic option, lncRNA signatures already represent a promising tool for personalized treatment and clinical management for individual patients.

## Figures and Tables

**Figure 1 ijms-24-01828-f001:**
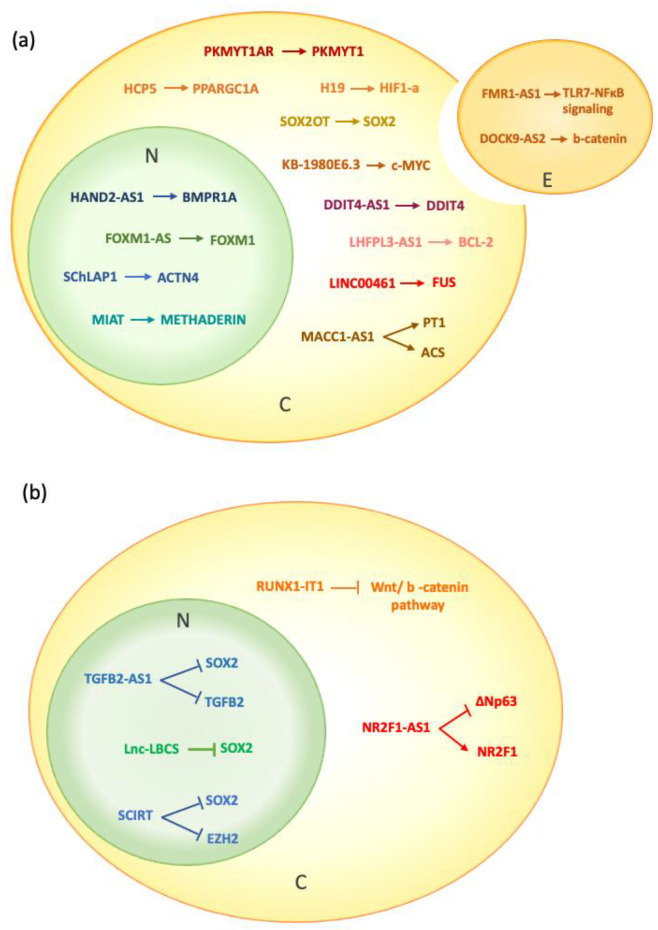
Schematic representation of lncRNAs localization and targets (**a**) lncRNAs that increase self-renewal and maintenance of CSCs. (**b**) lncRNAs counteracting the stemness of CSCs. N: nucleus; C: cytoplasm; E: exosome.

**Table 1 ijms-24-01828-t001:** LncRNAs working in the nucleus, through the interaction with epigenetic or transcription factors.

lncRNA	Modulation Tumor Type	Molecular Pathway	Role in Cancer	Ref.
**HAND2-AS1**	Up Liver cancer	INO80 complex on BMPR1A promoter	Oncogenic: increasing self-renewal and maintenance of liver CSCs.	[36]
**TGFB2-AS1**	Down Triple-negative breast cancer	Antagonizing SMARCA4 binding capacity on TGFB2 and SOX2 promoters	Tumor suppressor: inhibition of CSCs signaling	[37]
**SCIRT**	Up Breast cancer	EZH2 and SOX2 colocalization and binding	Tumor suppressor: repression of the transcriptional programs of self-renewal in CSCs	[38]
**lnc-LBCS**	Down bladder cancer	Scaffold to induce the formation of hnRNPK–EZH2 complex	Tumor suppressor: Inhibition of Self-Renewal and Chemoresistance of CSCs	[39]
**FOXM1-AS**	Up Glioblastoma	ALKBH5/demethylation of FOXM1 nascent transcripts and FOXM1 expression	Oncogenic: proliferation of glioblastoma stem cells and in vivo tumorigenesis	[40]
**SChLAP1**	Up Glioblastoma	Enhanced interaction between HNRNPL and ACTN4, increased stability of ACTN4 protein, promoted nuclear translocation of NF-kB	Oncogenic: increased proliferation of glioblastoma cells in vitro and in vivo	[41]
**MIAT**	Up Medulloblastoma	Metadherin/regulation of a set of microRNAs	Oncogenic: maintenance of the tumorigenic stem-like phenotype and resistance to radiotherapy	[42]
**DOCK9-AS2**	Up papillary thyroid carcinoma	Increasing β-catenin level trought interaction with Sp1 transcription factor	Oncogenic: restoring proliferation, migration, invasion, EMT, and stemness of PTC cells.	[43]

**Table 2 ijms-24-01828-t002:** LncRNAs working in the cytoplasm as competing endogenous RNAs and through the interaction with RBPs.

lncRNA	Modulation Tumor Type	Molecular Pathway	Role in Cancer	Ref.
**H19**	Up Breast cancer	Sponging let-7/up-regulation of HIF1-α	Oncogenic: increase in the glycolysis gatekeeper PDK1 under hypoxia	[56]
**RUNX1-IT1**	Down HCC	Sponging miR-632/ Up-regulation of GSK-3β	Tumor suppressor: repression of the Wnt/β -catenin pathway	[57]
**DOCK9-AS2**	Up PTC	Sponging miR-1972/ Up-regulation of CTNNB1	Oncogenic: enriched in the exosomes from PTC stem-like cells, which transfer this lncRNA to the non-stem tumor cells	[43]
**PKMYT1AR**	Up NSCLC	Sponging miR-485-5p/Up-regulation of PKMYT1	Oncogenic: activation of Wnt/β-catenin pathway	[58]
**SOX2OT**	Up Bladder cancer	Sponging miR-200c/Up-regulation of SOX2	Oncogenic: increase in stemness in vitro and tumorigenic and metastatic ability in vivo	[59]
**LHFPL3-AS1**	Up Melanoma	Sponging miR-181a-5p/Up-regulation of BCL2	Oncogenic: inhibition of the apoptosis of melanoma stem cells	[60]
**MACC1-AS1**	Up Gastric cancer	Sponging miR-145-5p/C Up-regulation of PT1 and ACS	Oncogenic: increases FAO-dependent stemness and chemoresistance	[61]
**HCP5**	Up Gastric cancer	SpongingmiR-3619-5p/Up-regulation of PPARGC1A	Oncogenic: increases FAO, stemness and chemoresistance	[62]
**KB-1980E6.3**	Up Breast cancer	Enhanced recognition of m^6^A-modified c-Myc mRNA via IGF2BP1, increased stability of c-Myc mRNA	Oncogenic: breast cancer cells stemness and tumorigenesis in vitro and in vivo	[63]
**DDIT4-AS1**	Up Pancreatic cancer	UPF1/increased DDIT4-AS1, promotion of UPF1 phosphorylation, degradation of DDIT4 mRNA	Oncogenic: activation of the mTOR pathway, increased stemness and reduced chemosensitivity	[64]
**NR2F1-AS1**	Higher in late vs early recurring tumors Breast cancer	PTBP1/enhanced translation of NR2F1, suppression of the transcription factor and marker of epithelial stem cells ΔNp63	Reduced metastatic proficiency. Promotion of the dormancy of breast cancer metastatic cells, EMT changes and impaired tumorigenicity	[65]
**LINC00461**	Up Glioblastoma	CNOT6, FUS/LINC00461 stabilization, competitive binding to miR-485-3p	Oncogenic: increased expression of key factors promoting cell division and proliferation	[66]
**FMR1-AS1**	Up Esophageal squamous cell carcinoma	Interaction with TLR7 to activate TLR7-NFκB signaling and c-Myc expression	Oncogenic: inducing ESCC cell proliferation and invasion ability	[67]

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
