# Peer review of "Long Noncoding RNAs and Cancer Stem Cells: Dangerous Liaisons Managing Cancer"

_ijms, 2023, doi:10.3390/ijms24031828_

Round 1

Reviewer 1 Report

Thanks for the comprehensive review on the long noncoding RNAs and cancer stem cells. The literature review is well-conducted, and the overall manuscript is well-written. However, several points need to be addressed before publication.

1. A review article about non-coding RNAs in cancer stem cells was published in 2018. DOI: 10.1016/j.canlet.2018.01.027. A review article about long non-coding RNAs in cancer stem cells was published in 2021. DOI: 10.1016/j.semcancer.2020.12.012. The author should highlight and discuss the differences and advantages of their review over these two reviews.

2. The figure quality is poor. Please increase the aesthetics of the Figure.

3. Epithelial-to-mesenchymal transition is an essential characteristic of cancer stem cells. The current description of lncRNA and EMT is insufficient.

4. The studies on long noncoding RNAs and cancer stem cells in cancer therapy should be discussed in-depth.

5. The author could find some relevant citations about the roles of lncRNA in cancer. "DOI: 10.1093/bib/bbac386" and "DOI: 10.7150/thno.74281"

Author Response

    1. A review article about non-coding RNAs in cancer stem cells was published in 2018. DOI: 10.1016/j.canlet.2018.01.027. A review article about long non-coding RNAs in cancer stem cells was published in 2021. DOI: 10.1016/j.semcancer.2020.12.012. The author should highlight and discuss the differences and advantages of their review over these two reviews.

    We carefully read the two reviews cited by the referee. The first review briefly described 7 nuclear lncRNAs involved in the maintenance of CSCs by epigenetic and transcriptional regulation. All the cited papers were published before 2018. The second one, focuses on the lncRNAs involved in the EMT transition. All described lncRNAs (most of which are described in papers published before 2018) are involved in EMT promotion and behave as inducers of CSC-like phenotypes.

    In our review we have chosen to analyze recent articles published in the last 5 years, focused on the impact of lncRNAs on CSCs regulatory mechanisms. We classified lncRNAs based on their localization (see paragraphs 3,4,5) and on their oncogenic/tumor suppressor properties (see tables and figure). The description of the lncRNAs that inhibit CSCs has allowed us to highlight how the same gene (e.g SOX2) in different contexts can be induced by some lncRNAs and repressed by others. We also showed that an up-regulated lncRNA does not necessarily play a role of inducer of stemness or pluripotency (e.g. SCIRT).

    We have added a sentence at the end of the Introduction where we highlight the peculiarity of our work compared to the existing reviews on the same topic.

    1. The figure quality is poor. Please increase the aesthetics of the Figure.

    We have increased the aesthetics of the figure as suggested by referee

    1. Epithelial-to-mesenchymal transition is an essential characteristic of cancer stem cells. The current description of lncRNA and EMT is insufficient.

     We thank the Referee for his/her suggestion and we added a short description of the link between CSCs and EMT in the introduction (page 1, line 39-41).  For what concerns the involvement of lncRNAs in EMT, we would like to underline that our review is set up to highlight the molecular functions of lncRNAs with respect to their localization. We mentioned EMT, as well as invasion, migration and stemness, when describing the role of RUNX1-IT1 (page 10) and NR2F1-AS1(page 13). Deepening the link between lncRNAs and EMT would necessarily lead to the deepening of other cellular aspects that have already been well described in other reviews. We believe that this goes beyond the aim and focus of our review.

    1. The studies on long noncoding RNAs and cancer stem cells in cancer therapy should be discussed in-depth.

    In the discussion, we have added a more extensive discussion of the therapeutic potential of lncRNA targeting in cancer, providing some very recent examples, including also the Referee’s suggestions (see below, point 5).

    1. The author could find some relevant citations about the roles of lncRNA in cancer. "DOI: 10.1093/bib/bbac386" and "DOI: 10.7150/thno.74281"

    We thank the Referee for his/her suggestion, that we employed in the present revised version of our Discussion.

Reviewer 2 Report

Ciafrè et al. summarized recent progress on how the lncRNAs get involved in the CSCs regulation. In this review, the CSCs-related lncRNAs are classified by their subcellular localization and divided into nuclear lncRNAs, cytoplasmic lncRNAs and exosomal lncRNAs. The authors pointed out that the subcellular localization of the lncRNAs is highly correlated with their molecular mechanisms and biological functions. The nuclear lncRNAs are usually involved in chromatin organization, transcription, and RNA processing. The cytoplasmic lncRNAs always play a role in regulating either miRNAs or RBPs. The major function of the exosomal lncRNAs is related to the extracellular communication. The authors listed several examples for each of these classifications and described in detail how these lncRNAs regulate the CSCs.

There are plenty of similar reviews published in the past several years in this area, including “Sonawala, K., et al., Cells, 2022”, “Schwerdtfeger, M., et al., Translational Oncology, 2021”, “McCabe, E. M., & Rasmussen, T. P., Seminars in cancer biology, 2021”, “Castro-Oropeza, R., et al., Cellular Oncology, 2018” and more. Some of them have much overlap with the manuscript and contain even more details. However, the manuscript could still be novel and meaningful due to the interesting classification of lncRNAs, different emphases and different lncRNA examples. To further improve the uniqueness of this review, the authors could highlight the most recent lncRNA findings not discussed in previous reviews. Overall, the manuscript is suitable for publication in Int. J. Mol. Sci., subject to the minor revisions described below.

1. As mentioned above, plenty of similar reviews have been published in the past several years. But these recent progresses were not cited in the manuscript. Although the manuscript has its uniqueness and different angle, I will suggest the authors could also refer to the other reviews in the field, which could make the review more professional. On the other hand, the authors could include more the most recent lncRNA examples published in the past 2-3 years, to move the field further ahead.

2. The authors like to use some complex grammar. This could be advantageous when expressing complete logic and explaining emerging nouns. However, some sentences are too long and complex which should be break down to two sentences. For example, in the sentence “Most … ncRNAs” on line 156-159 (page 4), there are two emerging concepts explained in a single sentence and make the referent of the “that” in the last part of the sentence unclear. I suggest the authors double check all the long sentences and avoid too complicated grammar.

3. Chapter 3 “Nuclear lncRNAs and CSCs” (page 3-5) is not properly paragraphed.

4. Figure 1 (page 12) should better distinguish the nucleus and cytoplasm by adding a circle.

Author Response

here are plenty of similar reviews published in the past several years in this area, including “Sonawala, K., et al., Cells, 2022”, “Schwerdtfeger, M., et al., Translational Oncology, 2021”, “McCabe, E. M., & Rasmussen, T. P., Seminars in cancer biology, 2021”, “Castro-Oropeza, R., et al., Cellular Oncology, 2018” and more. Some of them have much overlap with the manuscript and contain even more details. However, the manuscript could still be novel and meaningful due to the interesting classification of lncRNAs, different emphases and different lncRNA examples. To further improve the uniqueness of this review, the authors could highlight the most recent lncRNA findings not discussed in previous reviews. Overall, the manuscript is suitable for publication in Int. J. Mol. Sci., subject to the minor revisions described below.

  1. As mentioned above, plenty of similar reviews have been published in the past several years. But these recent progresses were not cited in the manuscript. Although the manuscript has its uniqueness and different angle, I will suggest the authors could also refer to the other reviews in the field, which could make the review more professional. On the other hand, the authors could include more the most recent lncRNA examples published in the past 2-3 years, to move the field further ahead.

We followed the Reviewer’s suggestion by citing other relevant review articles published in the last years. At the same time, by quoting those papers, we have also highlighted the uniqueness of our present work, focused on the subcellular localization of lncRNAs in cancer stem cells, and on the regulation of epigenetic and post-transcriptional states by these molecules. All papers discussed in our manuscript have been published in the last five years, thus providing an update of the state of the art in this field. In addition, in this revised version, we have inserted 5 more references (refs 85-87, 89, 90), all published in 2022, to deepen our discussion about lncRNAs in cancer therapy.

  1. The authors like to use some complex grammar. This could be advantageous when expressing complete logic and explaining emerging nouns. However, some sentences are too long and complex which should be break down to two sentences. For example, in the sentence “Most … ncRNAs” on line 156-159 (page 4), there are two emerging concepts explained in a single sentence and make the referent of the “that” in the last part of the sentence unclear. I suggest the authors double check all the long sentences and avoid too complicated grammar.

We thank the Reviewer for his/her comment. We have now checked our manuscript focusing on this aspect, and changed some complicated sentences.

  1. Chapter 3 “Nuclear lncRNAs and CSCs” (page 3-5) is not properly paragraphed.

We have amended this part.

  1. Figure 1 (page 12) should better distinguish the nucleus and cytoplasm by adding a circle.

We have followed the Referee’s suggestion, and improved the readability of Figure 1, by better distinguishing the nucleus from the cytoplasm, defining boundaries and using different colors to indicate different compartments.